# Overturning Children’s Misconceptions about Ruler Measurement: The Power of Disconfirming Evidence

**DOI:** 10.3390/jintelligence12070062

**Published:** 2024-06-22

**Authors:** Mee-Kyoung Kwon, Eliza Congdon, Raedy Ping, Susan C. Levine

**Affiliations:** 1Department of Child Studies, Seoul Women’s University, Seoul 01797, Republic of Korea; mkwon@swu.ac.kr; 2Department of Psychology, Williams College, Williamstown, MA 01267, USA; elc6@williams.edu; 3Center for Strategic Leadership Development and Social Innovation, Atlanta, GA 30338, USA; raedyping@gmail.com; 4Department of Psychology, University of Chicago, Chicago, IL 60637, USA

**Keywords:** cognitive development, education, mathematical development, linear measurement, spatial cognition, disconfirming evidence, structural alignment, analogy

## Abstract

Children have persistent difficulty with foundational measurement concepts, which may be linked to the instruction they receive. Here, we focus on testing various ways to support their understanding that rulers comprise spatial interval units. We examined whether evidence-based learning tools—disconfirming evidence and/or structural alignment—enhance their understanding of ruler units. Disconfirming evidence, in this context, involves having children count the spatial interval units under an object that is not aligned with the origin of a ruler. Structural alignment, in this context, involves highlighting what a ruler unit is by overlaying plastic unit chips on top of ruler units when an object is aligned with the origin of a ruler. In three experiments employing a pre-test/training/post-test design, a total of 120 second graders were randomly assigned to one of six training conditions (two training conditions per experiment). The training conditions included different evidence-based learning principles or “business-as-usual” instruction (control), with equal allocation to each (*N* = 20 for each condition). In each experiment, children who did not perform above chance level on the pre-test were selected to continue with training, which resulted in a total of 88 students for the analysis of improvement. The children showed significant improvement in training conditions that included disconfirming evidence, but not in the structural alignment or control conditions. However, an exploratory analysis suggests that improvement occurred more rapidly and was retained better when structural alignment was combined with disconfirming evidence compared to disconfirming evidence alone.

## 1. Introduction

Understanding measurement is critical in mathematics and science as well as in everyday life tasks including cooking, assembling furniture, and understanding maps, models, and graphs. However, children consistently struggle to understand key measurement concepts more than other foundational mathematical topics (National Assessment of Educational Progress [NAEP] 1996, 2003; for details, see [28]; [26]; [27]; [48]; [49]). Children’s misunderstanding of ruler measurement is particularly apparent when they are asked questions about the length of objects that are shifted to the right of the zero-point on the ruler (see Figure 1 for examples of both traditional and shifted problems). These difficulties persist into middle school (e.g., [2]), with more than three quarters of third-grade students and more than a third of eighth-grade students responding incorrectly to “shifted” ruler problems (86% incorrect responses by Grade 3 students; 78% Grade 4; 51% Grade 7; and 37% Grade 8; Carpenter et al. 1988; Lindquist and Kouba 1989). Thus, it is important to understand why children show these difficulties and what can be done to improve their understanding. Here, we carry out a set of studies to examine whether cognitive supports related to powerful cognitive learning principles are effective in improving second-graders’ understanding of a particular measurement concept—that rulers are composed of spatial interval units and that counting these units can reveal the length of an object.

Ruler measurement is children’s earliest exposure to the formal notion of “units of measure” and it typically first appears in kindergarten curricula. The National Council of Teachers of Mathematics Standards ([29]) and the Common Core State Standards in Mathematics ([6]) recommend that by the end of second grade, children should demonstrate an understanding of the unit concept through the mastery of more advanced measurement skills like using linear units to compare lengths of objects and to represent the magnitudes of numbers on a number line. We focus on second-grade students in our study because the Common Core Standards indicate that measuring with conventional instruments such as the ruler is a focus of math instruction starting in first grade and continuing into second grade ([6]).

To understand young children’s difficulty with ruler measurement, it is useful to consider the typical measurement activities they are exposed to in the classroom. A review of how measurement is taught in kindergarten through Grade 3 in three commonly used mathematics curricula used in the United States (Everyday Mathematics, third edition, [47]; Mathematics, Michigan edition, [4]; Saxon Math, [21]) revealed some common shortcomings with respect to measurement activities at these grade levels, despite the fact that these curricula were each written using different frameworks ([43]). This review revealed that instructional activities in kindergarten through third grade focus on measuring objects with non-conventional units (e.g., paper clips) *or* with rulers (but rarely in conjunction), with the use of the former decreasing across grades and the use of the latter increasing ([43]). The vast majority of instructional activities studied (75%) focused on the procedures that are used to measure rather than on the conceptual foundations of these procedures. For example, for activities with paper clips, the focus was on not leaving spaces or letting the paper clips overlap; for rulers, the focus was on aligning the object with the zero-point of the ruler and reading off the number on the ruler number that corresponds to the right-most edge of the object.

## 2. Structural Alignment

The curricula reviewed by [43] ([43]) dramatically under-utilize instructional activities that are likely to support measurement understanding, based on cognitive science research. One missed instructional opportunity is the absence of activities that encourage children to “structurally align” discrete units with the continuous ruler. Structural alignment—aligning the relational structures of different representations—facilitates comprehension by enabling learners to compare the understandings they have gleaned from separate instructional activities to gain new and deeper understandings ([11]). In the context of measurement, activities that aim to integrate common, but typically separate, measurement activities (e.g., presenting a ruler with discrete chips aligned simultaneously) might encourage students to understand how these activities conceptually connect. Providing concrete, countable unit chips that are structurally aligned with a ruler may help children make the crucial insight that a tool of continuous length (ruler) can be understood in terms of its discrete units (unit chips). In other words, children may gain a conceptual understanding that rulers consist of spatial interval units by connecting measuring objects with concrete items like chips to measuring with a ruler. More generally, prior research has shown that the structural alignment of relevant information supports learning by increasing awareness of the similarities and differences between representations that otherwise might not be noticed. The use of structural alignment has been shown to increase the learning of a variety of concepts by children and adults, including math problem solving and learning an engineering principle (e.g., [5]; [11]; [19]; [36]; [46]).

## 3. Disconfirming Evidence

The review of math curricula also reveals that there are very few opportunities for children to try to measure objects that are shifted with respect to the zero-point on the ruler, which may mean that their misunderstandings about units go unnoticed by educators. None of the curricula that were reviewed introduced “shifted object” problems until at least third grade. This is problematic because children’s strong performance on “unshifted” measurement problems—problems that are aligned with the zero-point on the ruler—likely reflect their shallow procedural knowledge of how to use the ruler, rather than their conceptual understanding of what constitutes a spatial unit (e.g., [1]; see Figure 1A). Arriving at “correct” answers on such problems is possible with a variety of strategies and/or conceptions about how rulers work as tools (see Figure 1A), but such strategies fail to provide educators with information about whether children are thinking about linear measurement correctly or incorrectly. The prevalent use of such problems reflects a “positive instruction bias”, analogous to individual learners’ positive test bias—the tendency to focus on test cases that are consistent with hypotheses (for a review, see [18]). That is, unshifted ruler problems tend to confirm incorrect or a lack of ideas about ruler units that, nonetheless, lead to correct answers.

In contrast to unshifted ruler problems, presenting children with a shifted ruler problem that consists of measuring an object not aligned with the zero-point on the ruler (e.g., placing its left edge above the 2-inch mark on the ruler) readily uncovers the misconceptions children have about ruler measurement. Shifted ruler measurement problems on standardized tests or in psychological experiments show that children commonly make one of two kinds of errors (e.g., [2]; [17]; [22]; [44]). One kind of error, the “read-off” error, involves simply reading off the number on the ruler that is aligned with the right-most edge of the object, and failing to account for the fact that the object is shifted with respect to the zero-point on the ruler. Importantly, this procedure works well when objects are placed at the zero-point on the ruler (Figure 1A), but results in an erroneous response when the object is shifted with respect to the zero-point (in Figure 1B, this error would result in the answer “5”). The second kind of error, the “hatch mark counting” error, involves counting the hatch marks that correspond to the object being measured rather than the spatial interval units, resulting in the correct answer when objects are placed at the zero-point on the ruler (Figure 1A), but a response that is one greater than the correct answer when the object is shifted with respect to the zero-point (in Figure 1B, the hatch mark counting answer would be “4”). This error may reflect children’s propensity to count discrete physical objects ([39]). 

Another benefit of providing children with shifted measurement problems is that they could provide a particularly powerful context for instructor feedback. For example, if a child sees an object that starts at the “2” on a ruler and ends at the “5”, they may confidently declare that the object is 5 units long. At this point, if a teacher corrects the child and offers the answer of “3”, they are providing an answer that directly disconfirms the child’s belief. Prior research has shown that disconfirming evidence after a self-generated, incorrect prediction is a strong driver of learning across a broad range of ages and content areas (e.g., [33]; [35]; [50]). As early as infancy, violations of expectation not only reveal early beliefs, but also increase exploration and the learning of new information. In a series of studies, [45] ([45]) found that when an object violated infants’ expectations about the physical world (e.g., by passing through a wall), they were more likely to explore that object and also to learn to associate a sound with that object compared to the same measures involving an object that did not violate an expectation. Such findings align with Bayesian views that learning occurs by weighing new evidence and prior beliefs ([37] for review). Consistent with the power of disconfirming evidence in supporting learning, instructional experiences that incorporate positive and negative examples of a concept result in more learning of a variety of concepts than positive evidence alone ([16]; [51]). In the context of linear measurement, activities utilizing shifted ruler problems serve as powerful instances of employing “disconfirming evidence” because children’s predictions do not match the answer they find when they count units with an instructor. In this way, shifted-object problems likely challenge a learner’s prior knowledge and beliefs to help drive conceptual change as the learner is forced to confront and reconcile contradictory information.

## 4. Current Study

In three experiments, we examine whether two kinds of ruler measurement instruction positively affect children’s conceptual understanding of ruler units. These activities involve (a) shifted ruler problems that children often respond to incorrectly (disconfirming evidence), and (b) aligning discrete chips with the continuous ruler, which provides children with the opportunity to compare these measurement activities and to more readily learn about the nature of ruler units (structural alignment). We predicted that both of these activities would increase children’s conceptual understanding that the units underlying ruler measurement are spatial intervals. Given prior studies (e.g., [3]; [28]; [25]; [26]; [27]; [43]; [44]; [48]), we used second-graders for our study to examine individuals who have already been introduced to ruler measurement as part of length measurement but still exhibit room for improvement in their understanding of measurement units.

While these learning principles (disconfirming evidence and structural alignment) do not represent an exhaustive list of ways to improve understanding of units, we selected them because they are evidence-based and potentially helpful to overturning children’s ruler misconceptions. In Experiment 1, we compare training with typical classroom measurement activities (business-as-usual or BAU) to training that involves both disconfirming evidence and the structural alignment of discrete unit chips with the ruler (disconfirming evidence + structural alignment or DE + SA). The activities in the business-as-usual condition resemble typical classroom measurement activities, which do not typically incorporate the principles of structural alignment and disconfirming evidence. These activities solely involve unshifted ruler tasks or informal measurement activities using familiar concrete objects (e.g., chips in our study). Presented as separate trials, they may fail to enable students to bridge their understanding of informal measurement with ruler measurement activities, thus lacking structural alignment. Furthermore, these activities may reinforce incorrect strategies among students as they do not provide an opportunity for them to observe that their original approach can yield incorrect results, indicating a lack of disconfirming evidence. Therefore, we hypothesized that business-as-usual training would *not* lead to changes in children’s performance on shifted ruler problems, whereas training on shifted problems with disconfirming evidence and structural alignment (discrete units overlaid on the ruler) would lead to such changes. 

In Experiments 2 and 3, we probe whether one or both of the learning tools we utilize—disconfirming evidence and structural alignment—drive children’s improvement.

## 5. Experiment 1—Disconfirming Evidence and Structural Alignment (DE + SA) vs. Business-As-Usual (BAU)

### 5.1. Method

#### 5.1.1. Participants 

Forty second-grade students (19 boys and 21 girls; mean age = 8.10 years; *SD* = 0.35 years) from a private school in a large urban area were tested using a pre-test/training/post-test design. Our sample was drawn from a school population with high SES backgrounds (tuition at the school was USD 16,000, though significant aid was available) and was racially and ethnically diverse (39% White, 19% Multiracial, 19% Asian or Asian American, 9% Black or African American, 5% Latinx, and 1% Middle Eastern, Pacific Islander, or Native American, according to school statistics). 

The responses to shifted items on the pre-test made it possible to distinguish which strategy a child predominantly used (correct; counting hatch marks; read-off; other). If a child answered at least 5 of 8 shifted measurement problems in accordance with a particular strategy, he or she was categorized as having started with that strategy. This criterion was based on a binomial distribution with 0.25 probability of choosing each alternative on each test item. Using this distribution and probability, children who answered 5 or more of the 8 problems using a particular strategy were statistically above chance to be relying on that strategy (the two-tailed *p*-value for X ≥ 5 was 0.027). One child was excluded because they did not consistently use a single pre-test strategy and eleven children were excluded because their primary pre-test strategy was “correct” and, thus, there was little or no room for improvement after training. The final sample for analysis consisted of 28 children (*N* = 13 males). Nineteen of these children used the hatch mark counting strategy at pre-test and nine used the read-off strategy at pre-test. The children using each of these strategies were assigned in alternation to each of the two training conditions. This resulted in 15 children (10 who used the hatch mark strategy at pre-test and 5 who used the read-off strategy) ending up in the experimental condition (disconfirming evidence + structural alignment) and 13 children (9 who used the hatch mark strategy and 4 who used the read-off strategy) ending up in the business-as-usual condition. 

#### 5.1.2. Testing Procedures and Materials

The students participated individually in a quiet area of their school. All participants completed the pre-test during the first session. During the second session, which occurred a week later, children received their assigned training condition as well as an immediate post-test. One week after that, they received a follow-up test. 

##### Pre-Test, Post-Test, and Follow-Up Test 

The pre-test, post-test, and follow-up test each consisted of 16 multiple-choice measurement questions. On each page, an image of a crayon was positioned above an image of a ruler, and the experimenter asked, “How many units long is this crayon?” To respond, the children indicated one of the four choices printed below the ruler. At each test point, each child answered 8 unshifted problems followed by 8 shifted problems. Alongside our primary focus on shifted problems, we included unshifted problems to mimic more traditional ruler measurement questions. This allowed us to compare the children’s answers to the two types of problems and to confirm our hypothesis that children would perform very well on the unshifted problems but not on the shifted problems. No feedback was given during this task on any problems at any of the time points. 

On each problem, the crayon’s left edge was depicted lined up with either the zero-point of the ruler (unshifted item) or with the 2-inch, 3-inch, or 4-inch points of the ruler (shifted item). For each shifted item, the incorrect numerical foils were strategic lures: one corresponded to the “read-off strategy” answer, and another corresponded to the “hatch mark counting” answer. The third foil choice was a number randomly selected from 1 to 8 that was different from the other three choices. The crayons varied in horizontal length from 2 to 6 units and the ruler itself was 9 units long. We utilized two question sets, Sets A and B, distributing them to participants in each condition in alternating sequences (A-B-A or B-A-B) for the pre-test, post-test, and follow-up test to minimize potential practice effects. We conducted our primary analyses by combining data from both sequences due to the absence of any clear order effects.

##### Training 

To begin the training session, regardless of condition, each child was introduced to the materials in the following way:

“This is a ruler. And each of these [*Experimenter points to the unit chips*] is one unit. They are all the same length. I also have sticks that are different colors and sizes [*Experimenter points to the sticks*]. We are going to play a game by measuring these sticks with our ruler and units.”

Following this introduction, each child measured 8 wooden sticks of 4 different lengths (2, 3, 5, 7 units long). In the experimental condition (disconfirming evidence + structural alignment), the experimenter placed one of the sticks just above the ruler, shifted away from the start point, and asked the child how many units long the stick was. The experimenter then asked the child to check the answer by using their unit chips. To do so, the child placed transparent unit chips on top of the units of the ruler. The experimenter again asked the child how many units long the stick was, then moved the stick to the start-point of the ruler. The child placed the chips on the part of the ruler that was aligned with the stick in its new position. This allowed the child to compare the measurements obtained in the shifted position to that obtained when the stick was aligned with the start-point of the ruler, and to see that they were the same. After each trial, the experiment provided feedback by stating, “See, the stick is N units long” regardless of whether the child provided a correct answer. In contrast to the experimental condition utilizing shifted ruler problems, as exemplars of “disconfirming evidence”, the business-as-usual (BAU) condition used unshifted ruler problems. As depicted in Figure 2A, left, each child received eight training trials that involved measuring sticks that were lined up with the zero-point on a ruler followed by eight trials that involved measuring sticks with unit chips, with no ruler present. The two types of activities were carried out separately, thus lacking structural alignment, as is typically the case in school, rather than in an integrated manner, as in the experimental condition. The rest of the training procedure was identical to that of the experimental condition. To minimize unexpected effects from the experimenter, the same experimenter conducted both training conditions with the children in the respective conditions in alternation. 

### 5.2. Results

#### 5.2.1. Performance before and after Training

When the object being measured is lined up with the zero-point on the ruler, multiple strategies—counting spatial interval units, counting hatch marks, or reading off the number on the ruler—result in the correct answer. Accordingly, performance on the eight unshifted measurement problems was nearly perfect at all three time points across both experiments with little to no variance (e.g., pre-test proportion correct *M* = 1.00 (*SD* = 0.00); post-test proportion correct *M* = 0.98 (*SD* = 0.08); follow-up proportion correct *M* = 1.00 (*SD* = 0.00)). As emphasized earlier, assessing performance on these problems alone, as is typical in elementary school classrooms, would leave teachers with the impression that children have very few, if any, misconceptions about ruler measurement. Children’s performance on the shifted ruler problems, however, reveals an entirely different picture. Thus, our analyses for both of our experiments focus on children’s performance on the eight shifted ruler problems only. 

To measure the relative effectiveness of each training condition on the shifted problems, we used a mixed-effects logit model to predict the odds of a correct response as a function of time point (pre-test, post-test, follow-up), training condition (DE + SA, BAU), and interaction between the two (time point x condition). This type of model was selected because our outcome variable (score out of 8 on the measurement assessment) was bimodal and, therefore, non-normally distributed with peaks at 0 and 8. In addition to the fixed effects described above, the model also had a random effect of participant to control for the fact that each participant contributed multiple data points at each session. 

An analysis of variance of the model revealed a main effect of time point (χ^2^ = 17.79, *p* = 0.00014), which was qualified by a significant two-way interaction between training condition and time point (χ^2^ = 40.19, *p* < 0.0001)[note 1]. There was no independent main effect of condition in this model (χ^2^ = 1.96, *p* = 0.16). To follow up on the condition by time point interaction, we built three separate models, one for each testing session. At pre-test, the groups did not differ (*β* = 0.78, *SE* = 0.82, *Z* = 0.96, *p* = 0.338), but after training, the children in the experimental condition (DE + SA) significantly outperformed the children in the BAU control group (*β* = 11.27, *SE* = 5.03, *Z* = 2.24, *p* = 0.025). This pattern was even more extreme at the follow-up session (*β* = 22.65, *SE* = 5.04, *Z* = 4.51, *p* < 0.00001). See Figure 3A for the average percentage of correct answers for each condition at each of the three testing sessions. 

#### 5.2.2. Exploratory Analysis—The Role of the Pre-Test Strategy

In conducting the main analysis, we observed anecdotally that children who began the study by using the read-off strategy seemed to learn much less from instruction overall than their peers who began by using the hatch mark counting strategy. While adding a three-way interaction term (Session X Condition X Strategy) does significantly improve the fit of the model over and above a model that does not account for starting strategy at all (χ^2^ = 33.51, *p* < 0.0001)[note 2], the interaction term itself is not statistically significant (χ^2^ = 0.04, *p* = 0.98), likely because so much of the variance is cleanly accounted for by the lower-order terms in the model. Consistent with this possibility, there was a very strong main effect of strategy in this study, whereby children who used the hatch mark strategy had significantly higher odds of answering questions correctly than children who used the read-off strategy (χ^2^ = 25.44, *p* < 0.000001). In addition, the main effect of Condition remained strong in this model (χ^2^ = 7.43, *p* = 0.006), as did the main effect of Session (χ^2^ = 18.72, *p* < 0.00001) and the interaction between Session and Condition (χ^2^ = 42.56, *p* < 0.0000000001). See Figure 3B for performance at all three time points broken down by strategy group (hatch mark counters compared to read-off strategy users at pre-test). 

Descriptively, we also observed that one of the five children who used the read-off strategy at pre-test and who received the experimental training condition used the hatch mark counting strategy at post-test and follow-up, whereas none of the four children who used the read-off strategy at pre-test in the business-as-usual training changed strategy at either post-test or follow-up. In contrast to this change in strategy, no child in either training condition switched from using the hatch mark counting strategy to using the read-off strategy.

### 5.3. Discussion

In Experiment 1, we found that a brief training session that involved disconfirming evidence *and* the alignment of unit chips with the ruler significantly improved performance at an immediate post-test and at a one-week follow-up. We also found evidence that children whose starting strategy was to simply read off the right-most number performed significantly worse overall than their hatch-mark-counting peers. In contrast, we found that the business-as-usual treatment, in which children never received disconfirming evidence and did not see chips aligned with the stick that was to be measured with the ruler, did not improve performance on this linear measurement task. These findings demonstrate that children do benefit from instruction that is enhanced by the use of disconfirming evidence and alignment of unit chips with the ruler on shifted ruler problems. Our second experiment was designed to probe whether employing *both* of these cognitive tools in combination is necessary for such improvement, or whether each of these tools on its own drives learning. 

## 6. Experiment 2—Disconfirming Evidence (DE) versus Structural Alignment (SA)

Experiment 1 established that instruction incorporating disconfirming evidence and structural alignment improved children’s performance on shifted ruler problems. In Experiment 2, we compared training with disconfirming evidence provided by shifted ruler problems to training with structural alignment of the chips on unshifted ruler problems. As in Experiment 1, the outcome measure of interest was performance on shifted ruler problems directly after instruction (post-test), and at a 1-week follow-up. 

### 6.1. Method

#### 6.1.1. Participants

Forty second-grade students (18 boys and 22 girls; mean age = 8.00 years; *SD* = 0.35 years) were tested in Experiment 2. The participants were drawn from the same private school population as Experiment 1, which was high SES and racially and ethnically diverse. Using the same criteria as outlined in Experiment 1, the children were categorized according to their predominant pre-test strategy (5 of 8 trials answered using a particular strategy). Our final sample, excluding the one child whose pre-test strategy could not be categorized and the 10 children who were already successful on shifted problems at pre-test, included 28 children (*N* = 13 males). Children using each of these strategies were assigned in alternation to each of the two training conditions. With this method, 14 children were randomized into the disconfirming evidence condition that involved shifted ruler problems and no chips (10 hatch mark counters and 4 read-off strategy users) and 14 children were assigned to the structural alignment condition that involved unshifted ruler problems with chips (9 hatch mark counters and 5 read-off strategy users). 

#### 6.1.2. Materials and Procedure

All aspects of the procedure were identical to Experiment 1 aside from the training portion. During training, the children received one of the two instructional conditions depicted in Figure 2B. In the structural alignment condition, each child received eight trials in which each stick started at the zero-point of the ruler. The child was asked how many units long the stick was, and after they answered, was asked to place the unit chips on top of the ruler to see if their answer was correct. In the disconfirming evidence condition, the child received eight trials in which each stick was moved away from the zero-point on the ruler (2, 3, or 4 units away) and asked how many units long the stick was. After the child answered, the experimenter moved the stick to the zero-point of the ruler and again asked how many units long the stick was. In both training conditions, the children were then given feedback through the experimenter’s statement, “See, the stick is N units long”, regardless of whether the child had produced a correct response on that trial.

### 6.2. Results

#### 6.2.1. Performance before and after Training

Following the same procedure as in Experiment 1, we used a mixed-effects logit model to predict the odds of a correct response as a function of time point (pre-test, post-test, follow-up), training condition (disconfirming evidence + structural alignment, business-as-usual), and the interaction between the two (time point x condition) as fixed effects. There was also a random effect of participant to control for the fact that each participant contributed multiple data points at each session. 

An analysis of variance of the model revealed a main effect of time point (χ^2^ = 21.86, *p* < 0.00001), a main effect of training condition (χ^2^ = 4.31, *p* = 0.038), and a significant two-way interaction between training condition and time point (χ^2^ = 38.51, *p* < 0.001)[note 3]. To follow-up on this interaction, we built three separate models, one for each testing session. At pre-test, the groups did not differ (*β* = 1.14, *SE* = 1.05, *Z* = 1.089, *p* = 0.276), but after training, children in the DE condition significantly outperformed children in the SA condition (*β* = 6.96, *SE* = 2.18, *Z* = 3.18, *p* = 0.0014). This pattern held steady at the follow-up session (*β* = 6.138, *SE* = 2.52, *Z* = 2.44, *p* = 0.015). See Figure 4A for average percent correct for each condition at each of the three testing sessions. 

#### 6.2.2. Exploratory Analysis—The Role of the Pre-Test Strategy

In light of our findings from Experiment 1 in which the starting strategy was a significant predictor of overall performance, we opted to rerun our model, adding pre-test strategy as a fixed effect. This analysis provided a significantly better fit for the data than the model that did not account for pre-test strategy (χ^2^ = 9.34, *p* = 0.025). The main effects of condition and time point and their interaction remained significant and there was a marginal main effect of pre-test strategy (χ^2^ = 2.97, *p* = 0.08) whereby children who started the study using the hatch mark counting strategy performed better overall than children who started with the read-off strategy (Figure 4B)[note 4]. 

When we examined whether or not the participants shifted in their use of incorrect strategies, we found that one of the four children using the read-off strategy at pre-test who received the DE training used the hatch mark counting strategy at post-test and one of the four in this condition (a different child) used the hatch mark strategy at follow up testing. In contrast, no child in either training condition switched from using the hatch mark counting strategy to using the read-off strategy.

### 6.3. Discussion

In Experiment 2, we found that the children’s performance improved when they were trained to measure objects shifted away from the zero-point on a ruler even when the training did not involve overlaying discrete unit chips on the ruler. These findings suggest that providing children with concrete evidence that directly disconfirms their initial response on a ruler measurement task is particularly important in changing their understanding of linear measurement, and aligning discrete unit chips on unshifted ruler problems is not sufficient on its own. When the children did not receive disconfirming evidence, as was the case for the structural alignment condition, they did not seem to re-evaluate and revise their initial strategies. This is likely because, with unshifted ruler problems, which are the kind of problems children typically encounter in school, counting the structurally aligned unit chips corroborates the answer children provide when they use a read-off strategy or a hatch mark counting strategy. 

Our exploratory analysis suggests that the training condition effect was primarily driven by children who used the hatch mark counting strategy at pre-test, though there was not a significant interaction involving pre-test strategy and timepoint, likely because we did not power the study to examine the effects of the pre-test strategy. Although this finding is consistent with our previous research, it is in need of replication in the context of the particular training conditions we used in this study ([7]).

## 7. Experiment 3—Disconfirming Evidence Alone (DE) vs. Disconfirming Evidence + Structural Alignment (DE + SA)

In Experiment 3, we directly compared training with and without chips on shifted ruler problems in order to determine whether adding unit chips to shifted problems would confer any additional benefit to improving children’s understanding of ruler measurement. In other words, we compared whether training that involved both disconfirming evidence and alignment (successful condition in Experiment 1) would result in more improvement than disconfirming evidence alone (successful condition in Experiment 2). A post hoc power analysis confirmed that this study was slightly underpowered for the effect size we were hoping to detect. Thus, the findings from Experiment 3 should be considered preliminary and null findings should not be overinterpreted (see footnote 6). 

### 7.1. Methods

#### 7.1.1. Participants

Forty second-grade children (18 boys and 22 girls; mean age = 8.00 years; *SD* = 0.40 years) were tested in Experiment 3. Using the same criteria for determining pre-test strategy as described in Experiments 1 and 2, we excluded 8 children from the analysis because they used a correct strategy at pre-test (they each got at least 5 of 8 pre-test trials correct). Thus, our final sample included 32 children (*N* = 17 males). A total of 17 participants were randomized into the DE + SA condition (11 hatch-mark-counters; 6 read-off strategy users) and 15 participants were randomized into the DE alone condition (8 hatch-mark-counters; 7 read-off strategy users). 

#### 7.1.2. Materials and Procedure

The materials used in Experiment 3 were identical to those used in Experiments 1 and 2. The procedure for the DE + SA condition (shifted/chips; see Figure 2C, left) and the procedure for the DE alone condition (shifted/no chips; see Figure 2C, right) were identical to the conditions in Experiment 1 and in Experiment 2, with the following exception: in both conditions, the experimenter only provided one phase of disconfirming evidence instruction. That is, the stick was not moved to the start point after the child responded[note 5]. Instead, the experimenter merely provided the following verbal feedback: If the response was correct, the experimenter said, “Yes, you’re right. The stick is N units long.”; if the response was not correct, the experimenter said, “Actually, the stick is N units long”. This methodological decision was made because, anecdotally, the experimenters had observed that children were revising their initial incorrect guesses after only one phase of training (with the object shifted) and did not seem to gain additional insight from moving the stick to the origin point of the ruler. Future work could investigate whether there are nuanced differences in long term outcomes from including the second step, though none were observed in the current study.

### 7.2. Results

#### 7.2.1. Performance before and after Training

Following the same procedure as Experiments 1 and 2, we used a mixed-effects logit model to predict the odds of a correct response as a function of time point (pre-test, post-test, follow-up), training condition (disconfirming evidence + structural alignment, disconfirming evidence alone), and the interaction between the two (time point x condition) as fixed effects. There was also a random effect of participant to control for the fact that each participant contributed multiple data points at each session. 

An analysis of variance of the model revealed a clear main effect of time point (χ^2^ = 114.57, *p* < 0.00001) but no main effect of training condition (χ^2^ = 0.93, *p* = 0.33). However, there was a significant two-way interaction between training condition and time point (χ^2^ = 8.05, *p* = 0.018)[note 6]. To follow up on this interaction, we built two separate models, one for each training condition. For the DE + SA training group, the children’s performance improved dramatically from pre-test to post-test (*β* = 4.86, *SE* = 0.60, *Z* = 8.15, *p* < 0.00001) and then was maintained from the post-test to the follow-up session (*β* = −0.35, *SE* = 0.48, *Z* = −0.71, *p* = 0.47). By contrast, the children in the DE alone group showed dramatic improvements at post-test (*β* = 6.42, *SE* = 0.97, *Z* = 6.51, *p* < 0.0001), but then their performance dropped off significantly from the post-test to the follow-up session (*β* = −2.41, *SE* = 0.56, *Z* = −4.33, *p* < 0.0001). In other words, the children in the DE alone condition did not retain their learning as well as those in the DE + SA condition. Nonetheless, there was not a significant difference by condition within any of the sessions (all *p*-values > 0.32), though it is possible that a significant difference between the groups at the follow-up session may have emerged with a larger sample size. See Figure 5A for the average number of problems answered correctly in each condition at each of the three testing sessions. 

#### 7.2.2. Exploratory Analysis—The Role of the Pre-Test Strategy

Once again, we reran our model, adding pre-test strategy as a fixed effect. This analysis provided a significantly better fit for the data than the model that did not account for the pre-test strategy (χ^2^ = 8.67, *p* = .003). As in our primary analysis, the effect of condition was not significant, the interaction between session and condition remained significant, and there was a significant main effect of pre-test strategy (χ^2^ = 8.03, *p* = .005) whereby children who started the study using the hatch mark counting strategy did significantly better overall than children who started with the read-off strategy (Figure 5B)[note 7]. 

When we examined whether or not the participants shifted in their use of incorrect strategies, we found that two of the seven children using the read-off strategy at pre-test who received DE alone training used the hatch mark counting strategy at post-test and one of those children continued to do so at follow-up testing. None of the six children using the read-off strategy at pre-test who received the DE + SA training switched to the hatch mark counting strategy at either post-test or follow-up. Further, as in Experiment 1 and 2, no child in either training condition switched from using the hatch mark counting strategy to using the read-off strategy.

## 8. Learning Trajectory *during* Training

Given the significant interaction between training condition and time point, we decided to examine whether there were differences in children’s learning *during* training in the two conditions tested. Specifically, we examined whether the DE + SA training would make the learning process more efficient compared to the DE alone condition. We measured the number of training trials that each child received before answering the first of at least three consecutive training trials correctly. For example, if a child responded incorrectly on the first training trial, correctly on the second training trial, incorrectly on the third training trial, and correctly on the fourth through sixth training trials, the number of trials to reach the first of three consecutive correct responses was coded as 4. If a child never produced three consecutive correct responses during training, the number of trials to reach three subsequent correct responses was coded as 8 (the maximum number of training trials).[note 8]

We conducted a two-way ANOVA on the mean number of trials needed to reach the criteria (means reported in Table 1). The main effects of condition (*F* (1, 27) = 3.571, *p* = 0.07) and pre-test strategy (*F* (1, 27) = 4.086, *p* = 0.053) were both marginally significant. The interaction effect of condition x starting strategy was not significant (*p* = 0.933). These results suggest that children who received the DE + SA training may have learned more quickly than those who received the DE alone condition, regardless of the starting strategy. In addition, there was a trend, consistent with the first two experiments, that children who counted hatch marks at pre-test learned faster than those who used the read-off strategy at pre-test, regardless of the training condition.

## 9. Effect of Condition across All Experiments

In order to better assess the potential impact of our training conditions with slightly more power, we conducted one final analysis that combined the data from all three experiments (See Figure 6 for the average number of correctly answered shifted ruler problems in combined analysis utilizing findings from Experiments 1, 2, and 3). We used a mixed-effects logit model to predict the odds of a correct response as a function of time point (pre-test, post-test, follow-up), training condition (DE + SA, DE alone, SA alone, and BAU), and interaction between the two (time point x condition). In addition to the fixed effects described above, the model also had a random effect of participant to control for the fact that each participant contributed multiple data points at each session. 

An analysis of variance of the model revealed a main effect of time point (χ^2^ = 144.79, *p* < .001) and a main effect of condition (χ^2^ = 8.19, *p* = .04), both of which were qualified by a significant two-way interaction between training condition and time point (χ^2^ = 109.28, *p* < .0001). To follow-up on the condition by time point interaction, we built three separate models, one for each testing session. At pre-test, the groups did not differ (χ^2^ = 2.63, *p* = 0.45), but at post-test, immediately after training, the children in the DE + SA condition showed significantly higher odds of a correct response than those in the SA alone condition (*β* = −6.82, *SE* = 1.99, *Z* = −3.43, *p* < 0.001) and the business-as-usual condition (*β* = −6.48, *SE* = 1.98, *Z* = −3.28, *p* < 0.01), but not the DE alone condition (*β* = 2.22, *SE* = 1.91, *Z* = 1.16, *p* = 0.24). In other words, the two conditions that provided disconfirming evidence were more effective than the two unshifted conditions, but they did not differ from each other at post-test. 

However, the story was different at the follow-up session. At this session, the odds of a correct response in the DE + SA condition were significantly higher than each of the other three conditions (*β*_DEalone_ = −7.26, *SE* = 3.52, *Z* = −2.06, *p* = 0.04); *β*_SAalone_ = −9.04, *SE* = 3.57, *Z* = −2.53, *p* = 0.01); *β*_BAU_ = −9.73, *SE* = 3.71, *Z* = −2.62, *p* = 0.009), whose estimates, in turn, did not differ statistically from one another. 

## 10. General Discussion

The current study used experimentally manipulated instruction to determine whether children’s understanding of linear measurement can be improved by using interventions based on research-supported learning principles. Our instruction tested two powerful learning tools from the cognitive science literature—disconfirming evidence and structural alignment—compared to a business-as-usual control condition as well as to each other. These tools were selected because they are evidence-based and intuitively seemed relevant in the context of linear measurement, where children must learn to inhibit common incorrect procedural strategies *and* adopt a novel, correct understanding of how units are represented on rulers. Four main findings emerged. First, we found that the business-as-usual condition, modeled after typical classroom instruction, failed to promote an understanding of ruler units in our sample. Second, we found that when children received training that involved measuring objects shifted away from the zero-point on the ruler, their likelihood of learning increased dramatically. Third, we found evidence that the structural alignment of discrete plastic unit chips with the ruler in combination with disconfirming evidence led to more retention of the learning seen at post-test and marginally increased the speed of learning during training compared to disconfirming evidence alone.

Our fourth main finding, although based on exploratory analyses, suggests that children who differed in terms of their pre-test knowledge and strategies responded differentially to training (see [8] for converging evidence). In particular, children who used the hatch mark counting strategy to solve shifted problems at pre-test were more likely to benefit from instruction overall than those who used the read-off strategy at pre-test. This finding supports the view that the hatch mark counting strategy is more sophisticated than the read-off strategy (e.g., [8]; [44]), and is consistent with prior findings that younger children tend to make read-off errors and older children tend to make hatch mark counting errors on shifted ruler problems ([22]; [44]). At the very least, the children who counted hatch marks understood something that the children who used the read-off strategy may not have understood—that the relevant ruler units are those that correspond to the spatial extent of the object and that these units need to be counted, albeit without understanding the nature of these units. It may also be the case that hatch-mark-counting children, compared to the children who use the read-off strategy, are less sure about the correctness of their strategy, and are therefore more ready to learn the correct strategy. This aligns with the view that children begin with naïve theories that are abstract, coherent representations of structure that support prediction, explanation, and counterfactual reasoning ([12]) and that the strength of these theories is integrated with new evidence to predict learning and belief revision ([38]). In the math domain, there is evidence that when children’s errors are based on misconceptions, they are particularly likely to have poor meta-cognitive monitoring skills and to be confident in their incorrect answers (e.g., [14]; [13] Forthcoming; [30]). As reported in these studies, this leads to change resistance, and makes it particularly important to utilize effective, evidence-based instructional strategies, a view that is corroborated by our findings on the kinds of instructional strategies that work to overturn ruler misconceptions. 

The finding that children who used the erroneous read-off strategy on shifted ruler problems did not learn very much overall from our interventions raises an important question—how can we advance the understanding of ruler units in children who are using the read-off strategy? In the current sample, very few children in the read-off group adopted a correct strategy after training, despite some previously published evidence that combining unit chips with a shifted-object problem *can* lead to improvement in this strategy group ([8]). One key difference between the current set of experiments and this work is that our sample was second-grade children rather than first-grade children, as in [8] ([8]). It is quite possible that children who persist in using the read-off strategy well into second grade are a different group than those who use the strategy in first grade, where it is more common and more developmentally appropriate. For example, it is possible that second-graders who persist in using the read-off strategy have lower inhibitory control than other students at their grade level, consistent with evidence that children with high inhibitory control are much more likely to adopt new, correct strategies on a linear measurement task than those with lower inhibitory control, even after controlling for other measures like working memory and general math skill ([34]). Taken together, these findings underscore the necessity of identifying and rectifying measurement misconceptions early, and of thinking critically about how to support learning when a child does show a more rigid adherence to misconceptions and incorrect procedures. 

One possibility is that second-grade children who use the read-off strategy merely require a higher dose of the instruction that was provided in the present study. In support of this possibility, there *were* three children across our experiments (*N* = 1 in Experiment 1; *N* = 2 in Experiment 2) who changed from the read-off strategy to the more advanced hatch mark counting strategy after training with shifted objects, showing some malleability in strategy use and perhaps some improvement in conceptual understanding. Further research is needed to determine whether this shift in strategy represents a beneficial intermediate stage in a protracted learning trajectory or whether read-off children have simply switched from one type of conceptual error to another. Nonetheless, our findings on learning differences based on strategies are valuable for educators, as students using both types of erroneous measurement strategies may coexist in the same mathematics class (e.g., [44]) and they clearly necessitate different interventions or, at the very least, different intervention dosages. Children using the read-off strategy may need a somewhat different approach, perhaps one that more explicitly calls their strategy into question by providing a different type of disconfirming evidence. For example, once children using the read-off strategy provide the answer “7” for a 3-inch stick that extends from the 4-inch mark on the ruler to the 7-inch mark, they could be shown a much longer 7-inch stick that extends from the beginning of the ruler to the 7-inch mark. Although the read-off strategy leads to the same numerical answer in both cases, the sticks are clearly different in length. Such experiences might call the read-off strategy into question and open the children to adopt the more sophisticated hatch mark counting strategy, and eventually even the correct strategy of counting spatial interval units. 

Returning to our first two key findings, one important contribution of the current study was that we identified several kinds of instruction that are ineffective for *all* learners. In Experiment 1, we found that the children did not improve after receiving the business-as-usual condition, which was modeled directly on typical classroom measurement activities. Similarly, in Experiment 2, we did not observe learning gains for any children who received aligned ruler problems, even when discrete unit chips were super-imposed on the ruler, a support we had predicted would be helpful on its own. These findings have direct implications for educational practice. As mentioned above, an examination of three widely used mathematics textbooks ([43]) revealed that a large majority of all length measurement content in Kindergarten through second-grade focused on procedural steps (i.e., how to measure with a ruler rather than what the answer means on a deeper conceptual level). The curricula used precisely the kinds of measurement problems that we found do not lead to learning—those that involved placing objects at the zero-point on the ruler. As mentioned previously, none of the three curricula that were analyzed included any shifted ruler problems until at least third grade. Lack of exposure to conceptually challenging measurement questions is problematic because children have no reason to think deeply about the nature of ruler units when they can use read-off or hatch mark counting strategies because these strategies reliably yield the correct answer to traditional unshifted ruler measurement problems. 

By contrast, our research findings provide compelling evidence that providing disconfirming evidence is a powerful instructional strategy for addressing linear measurement misconceptions in second-grade students. And Experiment 3, along with our exploratory combined analysis, suggests that combining these two learning principals (disconfirming evidence and structural alignment) might result in more rapid learning and longer-lasting learning about ruler measurement than disconfirming evidence alone. Why might this be the case? In terms of the greater speed of learning in the disconfirming evidence and structural alignment condition, one possibility is that disconfirming evidence alone provides children with feedback that their existing strategy is incorrect, but does not provide them with definitive evidence about what the correct strategy is. Given that we know that younger children—3- to 5-year-olds—have difficulty utilizing units of continuous quantity such as cupfuls to make magnitude judgments ([15]), it is not surprising that children in the DE alone condition took longer than children in the DE + SA condition to identify the counting of spatial interval units as the correct strategy. In other words, the DE + SA training condition may have helped children both reject their old strategies and to more quickly conceptualize spatial interval units as the relevant ruler units.

In terms of the longer-term effects of training, our examination of condition effects across three studies showed that the children in the DE + SA group were more likely to retain their learning over time than the children in the DE condition. The children in the DE + SA condition not only had access to the nature of spatial interval units through the alignment of chips on the shifted ruler problems as discussed above, but they were also more actively engaged in the training because they iteratively placed each unit chip on the ruler. Engaging the motor system through relevant actions or gestures has been shown to result in more robust and longer-lasting learning than comparable instruction without motor involvement on a variety of spatial and mathematical tasks (e.g., [10], [9]; [24]). Thus, the iterative placement of chips on the ruler could have contributed to the advantage of the DE + SA condition one week after training at the follow-up session.

In general, we know that children are not likely to change their old, incorrect strategies unless new strategies offer large advantages in terms of accuracy, speed, or a decrease in the number of processing steps ([23]; [40]; [41], [42]). A particularly exciting implication of the current study is that a thoughtfully designed instructional activity—one that helps the child discover a conflict with their own conceptual understanding—can promote rapid learning. It is striking that, after a brief training session (under ten minutes), we found evidence that a remarkably consistent and seemingly robust misconception about units could be overturned by providing a problem that incorporated disconfirming evidence. Moreover, the learning that was observed immediately after training was maintained one week later, suggesting that the conceptual change was not fleeting. In the context of linear measurement, we believe that these findings are likely to generalize to *any* child who uses the hatch mark counting strategy regardless of socio-economic, racial, or ethnic background (though prior findings suggest that the base rates of these strategies may differ by socio-economic status (e.g., [20])). Beyond the context of linear measurement, our findings raise the possibility that the careful construction of instructional activities that provide disconfirming evidence, particularly when combined with structural alignment, may be a powerful way to support rapid and long-lasting learning. 

## 11. Conclusions

Children have struggled with linear measurement for decades—the United States performed statistically significantly below the international average on measurement and geometry items in the most recently administered Trends in International Mathematics and Science Study ([49]). Internationally, only an average of 29% of fourth-grade children can correctly solve a challenging shifted measurement problem ([32]). And we know that understanding units of measure is crucial for success in mathematics and science, as it forms the foundation for comprehending discrete quantities, which are widely applicable across science, technology, engineering, and mathematics (STEM) disciplines ([31]). Yet, very few studies have systematically examined evidence-based instructional strategies for improving children’s understanding of measurement units. Our results show that typical ruler measurement problems are not effective, likely because they do not disconfirm—or even uncover—children’s most common misconceptions about the nature of linear units. We found that giving children a brief training session with shifted ruler items that disconfirm prior strategies led to significant increases in understanding that persisted at least one week later. Moreover, our exploratory findings suggest that this is particularly the case when shifted ruler problems include unit chips that are structurally aligned with the ruler and the to-be-measured object. This effect was driven by children who began the study by counting hatch marks on the ruler, which was the more common strategy among the second-grade students in our sample. 

These findings have practical implications for how measurement instruction could be modified to increase children’s understanding of linear measurement, which, in turn, may have downstream consequences for their understanding of units and measurement more generally. In addition, our findings illustrate the strength of using evidence-based learning principles to improve children’s learning of an important mathematical idea, and raise the possibility that utilizing these principles has the potential to enhance learning outcomes in mathematics and other academic domains more generally.

## Figures and Tables

**Figure 1 jintelligence-12-00062-f001:**
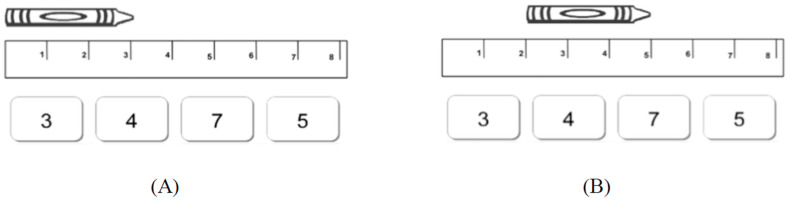
An example of an unshifted ruler problem (**A**) and a shifted ruler problem (**B**), both of which were used at pre-, post-, and follow-up testing. The correct answer for each of these problems is 3.

**Figure 2 jintelligence-12-00062-f002:**
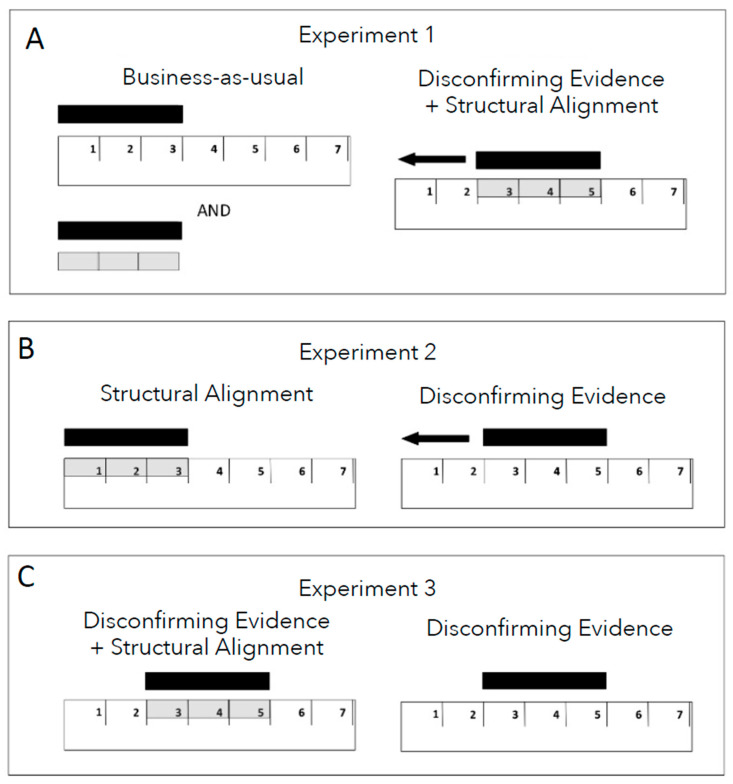
A schematic image of the training conditions for Experiments 1, 2, and 3 (**A**–**C**, respectively). In the experiments, children were provided with paper rulers, three-dimensional wooden rods to measure, and, when applicable, transparent plastic unit chips to overlay on the ruler or underneath the object.

**Figure 3 jintelligence-12-00062-f003:**
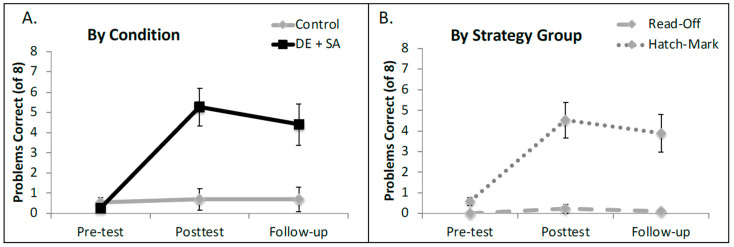
Mean number correct in terms of shifted ruler problems in Experiment 1. (**A**) shows the data broken down by training condition and (**B**) shows the data broken down by children’s dominant starting strategy at pre-text. Error bars represent ±1 standard error of the mean. DE = disconfirming evidence; SA = structural alignment; Control = business-as-usual.

**Figure 4 jintelligence-12-00062-f004:**
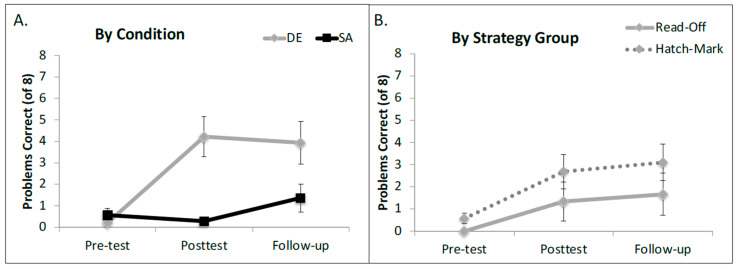
Mean number correct on shifted ruler problems in Experiment 2. (**A**) represents the significant main effect of training condition and (**B**) shows the marginal effect of starting strategy. Error bars represent ±1 standard error of the mean.

**Figure 5 jintelligence-12-00062-f005:**
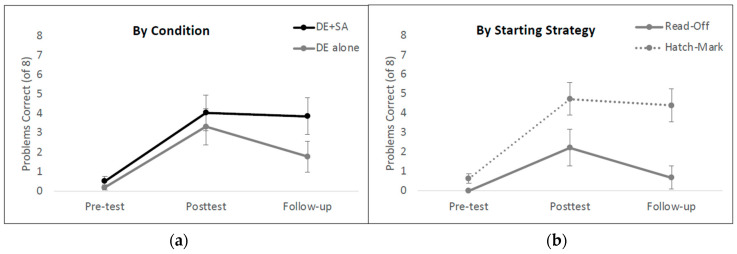
Mean number of correctly answered shifted ruler problems in Experiment 3. (**A**) represents the interaction between condition and session whereby children in the DE + EA condition maintain their learning, while those in the DE alone condition do not. (**B**) shows the significant main effect of starting strategy. Error bars represent ±1 standard error of the mean.

**Figure 6 jintelligence-12-00062-f006:**
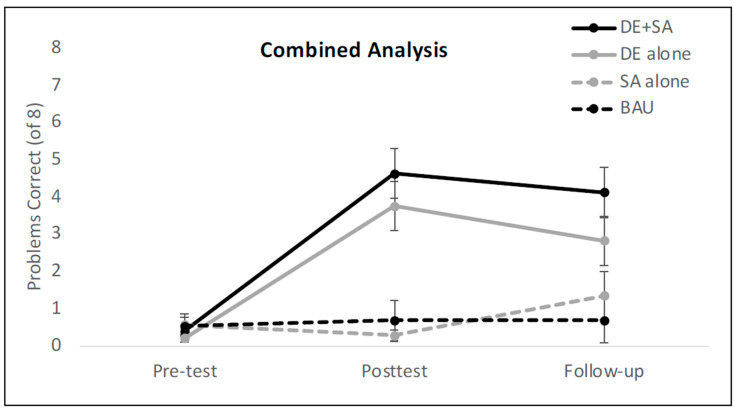
Mean number of correctly answered shifted ruler problems in combined analysis utilizing findings from Experiments 1, 2, and 3. The model revealed that children in the DE + SA condition maintained their learning at the follow-up session more so than those in the DE alone condition. Both groups outperformed the children in the SA alone and BAU conditions at post-test and follow-up. Error bars represent ±1 standard error of the mean. DE = disconfirming evidence; SA = structural alignment; BAU = business-as-usual.

**Table 1 jintelligence-12-00062-t001:** The average number of training trials before a child answered three questions correctly in a row (note that lower scores indicate *faster* learning).

Training Condition	Hatch Mark		Read-Off	
	Mean	SD	Mean	SD
Disconfirming Evidence Alone	2.88	2.80	4.71	1.34
Disconfirming Evidence + Structural Alignment	1.00	1.00	3.00	2.92

## Data Availability

The data presented in this study are available on request from the corresponding author, Susan C. Levine: s-levine@uchicago.edu.

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
