# Peer review of "Overturning Children’s Misconceptions about Ruler Measurement: The Power of Disconfirming Evidence"

_jintelligence, 2024, doi:10.3390/jintelligence12070062_

Round 1
Reviewer 1 Report
Comments and Suggestions for Authors
The authors wrote the manuscript, entitled “Overturning Children’s Misconceptions about Ruler Measurement: The Power of Disconfirming Evidence.” It is quite clear and easy to follow. Moreover, it is interesting and I am satisfied with the authors’ effort and good writing.
However, I have some suggestions for its improvement.
Abstract
· It would be better to add more information regarding the sampling technique and the total number of participants in the study, including each experimental study.
Introduction
The authors should give more information such as;
· What is the ‘disconfirming evidence’ in mathematics?
· Why is it important for children?
· What are the related findings regarding the disconfirming evidence?
· And the same, what is “the structural alignment?” Then, what is the related information regarding the research findings?
· What is the business-as-usual method? How much is it different from the above methods of ‘disconfirming evidence and structural alignment?
· The experimental figures showed these three methods. However, they should be matched with the theoretical descriptions of each method under the theoretical framework.
· What are the research gap, aims, and specific objectives/questions of this study?
· The authors combined all information such as the introduction, related reviews, and a theoretical framework for this study. Therefore, it is confusing to the readers. It is highly recommended that it would be clearer if the author divided all information into several sub-topics of related information. Especially, in this experimental study, the authors should give a clear theoretical framework.
Methods
In the procedure, please clarify some confusion such as;
· Please give more information, regarding “Pre-test, Post-test and Follow-up Test.” Are they the same? If they are the same, how did the authors prevent the potential impacts of pre-tests on the other two tests? Then, how much are they reliable?
· Were both the experimental group and the control group (the business-as-usual group) given the same training? By the same trainer?
· The authors should consider providing a rationale behind choosing Disconfirming evidence + Structural alignment as an experimental group and the reasoning behind the Business-as-usual approach in the control group.
· The feedback provided during the training session is explained. How did the authors consider the potential impact of this feedback on the learning process and how it might influence participants’ performance in subsequent tests?
· It is mentioned that random assignment was used, but details on the randomization procedure are not provided. Please consider including information on how randomization was achieved to ensure transparency and minimize bias.
· The description of measurement questions is clear. It might be beneficial to elaborate on the decision to include both unshifted and shifted problems, providing a rationale for this design choice.
(Please consider the above comments for all three experiments).
Findings
It is clear and easy to follow. However, please consider the following.
· What type of analysis is used to compare before/after the training and follow-up activities?
· The use of a mixed-effects logit model is appropriate. However, it would be beneficial to include some additional details on the model specifications, such as fixed and random effects, to enhance the transparency of the statistical approach.
· The decision to conduct separate models for each testing session is reasonable. However, it would be useful to elaborate on the significance of the observed pattern, especially the increasing difference in performance between the experimental and control groups over time.
· The observation of strategy change among children using the read-off strategy in the experimental training condition is well-noted. However, it would be better to discuss the potential reasons for this shift and its implications for instructional strategies.
· The inclusion of exploratory analysis in the pretest strategy is good. However, it would be clearer to discuss the practical implications of the observed differences in learning between children who started with the read-off strategy and those who began with the hatch-mark counting strategy.
Please consider the above comments for all three experiments.
General Discussion/Conclusion
The authors wrote this discussion very well. I suggest adding some information regarding the limitations of the study.
I have concerns about the suitability of this manuscript for the journal's aims and scope. The research just focuses on experimental findings related to the disconfirming evidence of ruler measurement in young children. What is the relevance of the findings for the readers of the Journal of Intelligence? Does the study fit well with the aims and scope of the journal?
Reviewer 2 Report
Comments and Suggestions for Authors
The authors presented three studies examining the effectiveness of different instructional methods aimed at enhancing children's understanding of ruler measurement units. The results showed that using disconfirming evidence as an instructional method significantly improved children's understanding of ruler measurement units. However, there was no significant improvement observed when employing structural alignment or a control condition. These findings highlight the potential effectiveness of disconfirming evidence in enhancing children's understanding of measurement concepts and suggest a possible synergistic effect when combining it with structural alignment. While the manuscript provides valuable insights, some aspects require more considerations.
(1) The manuscript would benefit from a more comprehensive discussion on the theoretical framework underpinning the chosen instructional methods. For example, elaborating on the cognitive processes/underpinnings involved and how these interventions align with current educational/cognitive theories would strengthen the theoretical basis of the study.
(2) More information on the selection criteria for 2nd graders would enhance the reader's understanding of the study's generalizability.
(3) I am wondering whether it is possible to compare the different conditions across the three experiments since different instructional methods were employed in each. Contrasting the differences among the three experiments could shed light on deeper underlying mechanisms. For instance, the DE+SA and DE conditions in Experiment 3 seem to lack the step of "moved the stick to the start-point of the ruler", which was present in Experiments 1 and 2. According to the authors' notes, this step seemingly did not contribute much. Such results are helpful in clarifying which instructional processes actually facilitate learning.
(4) It would be beneficial to provide annotations for the abbreviations (e.g., DE, SA) appearing in each figure.
Round 2
Reviewer 1 Report
Comments and Suggestions for Authors
The authors responded to the comments and suggestions very well. Now, I have no further comment. I think it is ready to be published.